# Warming accelerates belowground litter turnover in salt marshes – insights from a Tea Bag Index study

Hao Tang[1,2], Stefanie Nolte[3,4], Kai Jensen[2], Roy Rich[5], Julian Mittmann-Goetsch[2] and Peter Mueller[5,6]

[1]Key Laboratory of Land Resources Evaluation and Monitoring in Southwest, Ministry of Education, Sichuan Normal University, Chengdu, 610068, China
[2]Institute of Plant Science and Microbiology, Universität Hamburg, Hamburg, 22609, Germany
[3]School of Environmental Sciences, University of East Anglia, Norwich, NR47TJ, UK
[4]Centre for Environment, Fisheries and Aquaculture Science, Pakefield Rd, Lowestoft, UK
[5]Smithsonian Environmental Research Center, Edgewater, MD 21037, United States
[6]Institute of Landscape Ecology, Münster University, 48149 Münster, Germany

*Correspondence to*: Peter Mueller (mueller.p@uni-muenster.de)

## Abstract

Salt marshes play an important role in the global carbon (C) cycle due to the large amount of C stored in their soils. Soil C input in these coastal wetland ecosystems is strongly controlled by the plant primary production and initial decomposition rates of plant belowground biomass and litter. This study used a field warming experiment to investigate the response of belowground litter breakdown to rising temperature (+1.5°C and +3.0°C) across whole-soil profiles (0-60 cm soil depth) and the entire intertidal flooding gradient ranging from pioneer zone via low marsh to high marsh. We used standardized plant materials, following the Tea Bag Index approach, to assess the initial decomposition rate ($k$) and the stabilization factor ($S$) of labile organic matter (OM) inputs to the soil system. While $k$ describes the initial pace at which labile (= hydrolyzable) OM decomposes, $S$ describes the part of the labile fraction that does not decompose during deployment in the soil system and stabilizes due to biochemical transformation. We show that warming strongly increased $k$ consistently throughout the entire soil profile and across the entire flooding gradient, suggesting that warming effects on the initial decomposition rate of labile plant materials are independent of the soil aeration (i.e. redox) status. By contrast, negative effects on litter stabilization were less consistent. Specifically, warming effects on $S$ were restricted to the aerated topsoil in the frequently flooded pioneer zone, while the soil depth to which stabilization responded increased across the marsh elevation gradient via low to high marsh. These findings suggest that reducing soil conditions can suppress the response of belowground litter stabilization to rising temperature. In conclusion, our study demonstrates marked differences in the response of initial decomposition rate vs. stabilization of labile plant litter to rising temperature in salt marshes. We argue that these differences are strongly mediated by the soil redox status along flooding and soil-depth gradients.

**Keywords:** climate change, soil carbon, whole-soil profile, deep warming, blue carbon, tidal wetland, microbial activity

## 1. Introduction

Salt marshes provide a multitude of ecosystem services, such as wildlife conservation, flood protection, and water quality improvement (Barbier et al., 2011; Kirwan and Megonigal, 2013). Recently, salt marshes have additionally been recognized for their ability to store large amounts of organic carbon (C) in their soils, which has been acknowledged by the now common use of the term *blue carbon* (Chmura, 2013; McLeod et al., 2011). Global warming yields the potential to influence C

sequestration in salt marshes by affecting the balance between organic matter (OM) input to the soil system, through plant primary production, and output, through microbial decomposition of plant OM inputs (Kirwan and Mudd, 2012). Most of the current debate regarding the temperature sensitivity of the salt-marsh C balance is dealing with aboveground processes, i.e. plant primary production (Hamann et al., 2018; Kirwan and Blum, 2011; Liu et al., 2018; Noyce et al., 2019) and aboveground or surface litter breakdown (Charles and Dukes, 2009), whereas the effects on belowground processes remain largely unexplored (but see Mueller et al., 2018; Noyce et al., 2019). Belowground litter input and turnover often are more important drivers of soil C sequestration than aboveground processes (Kirwan and Megonigal, 2013; Langley and Megonigal, 2010), because aboveground litter gets deposited in an oxic environment and is subject to fast decomposition (Ozalp et al., 2007), whereas belowground litter can get stabilized under reducing soil conditions (Hatton et al., 2015; Lajtha et al., 2018; Poirier et al., 2018).

Litter decomposition dynamics are commonly quantified using litter-bag techniques. Litter bags are mesh bags filled with native plant litter of variable quality (e.g. with respect to C:N ratio or labile vs. recalcitrant fractions) that get deployed in the ecosystem or soil system in question. Initial decomposition rates are calculated based on litter mass loss over time. However, the mechanistic insight into climate change effects that can be gained from it is often limited owing to the challenge of separating climate from litter-quality effects (Prescott, 2010). Litter decomposition is controlled by complex interactions between litter-quality and climate parameters (Zhang et al., 2008). To separately assess climate effects on litter decomposition, it has proven useful to standardize litter quality. The Tea Bag Index *sensu* Keuskamp et al. (2013) represents a widely used standardized litter bag approach. It allows for the quantification of two key litter-breakdown parameters: the initial decomposition rate, $k$, and the stabilization factor, $S$.

Warming effects on litter breakdown in coastal wetlands may be strongly controlled by hydrology and resulting soil redox gradients. For instance, warming studies conducted in boreal peatlands demonstrated a dramatic reduction of warming effects on soil decomposition processes in waterlogged, and thus strongly reducing subsoils compared to less reducing topsoils (Hopple et al., 2020; Wilson et al., 2016). The authors suggest that the absence of oxygen can inhibit warming effects on soil microbial activity because phenolic compounds accumulate under anoxic conditions and inhibit microbial hydrolytic enzyme activity via the "enzymic latch" mechanism. In order to understand warming effects on litter breakdown in coastal wetland ecosystems, it is therefore necessary to use a standard substrate to study temperature effects across gradients in both flooding frequency (i.e. along the marsh elevation gradient) and in relation to soil depth.

The present study uses a unique field experiment to assess the effects of rising temperatures on litter breakdown in relation to soil depth and across the marsh elevation gradient ranging from pioneer zone via low marsh to high marsh. The MERIT (Marsh Ecosystem Response to Increased Temperature) experiment operates in a NW European salt marsh and induces active temperature manipulation to 1 m soil depth. We investigated litter breakdown for two consecutive growing seasons during the growing season of year 1 and year 2 of the experiment. We hypothesized (1) that warming will increase litter initial

decomposition rate, $k$, and decrease the ability of litter to stabilize, $S$, in soil, and (2) that warming effects on $k$ and $S$ will be inhibited by reducing soil conditions and thus, vary along gradients of both flooding frequency and soil depth.

## 2. Methods

### 2.1. Site description

The MERIT ecosystem warming experiment is located in a NW European salt marsh at Hamburger Hallig, Germany (54°36'06.2"N, 8°49'00.1"E) and has operated since spring 2018. The site is located on the coast of the Schleswig-Holstein Wadden Sea (Figure 1a and 1c) and has been protected as part of a National Park since 1985. This area is exposed to a temperate maritime climate; the annual mean temperature and mean precipitation are 10°C and 850 mm, respectively. The site is exposed to a tidal range of 3.4 m, with floodwater salinity ranging from 25 to 29 ppt (Mueller et al. 2023). The vegetation shows a zonation typical for Wadden Sea salt marshes: *Spartina anglica* is dominant in the pioneer zone, *Artiplex portulacoides* and *Puccinellia maritima* are dominant in the low marsh, and *Elymus athericus* is dominant in the high marsh (Esselink, 2017; Mueller et al., 2020a). The mean elevation of the pioneer zone, low marsh, and high marsh are 136 cm, 174 cm, and 212 cm (NHN, German standard ordnance datum), respectively. The pioneer zone is a typical feature of NW European salt marshes and is defined as the area where pioneer vegetation covers $\geqslant 5\%$ (Peterson et al., 2014). In the Wadden Sea region its average surface elevation is below mean high tide. Thus, the pioneer zone is typically flooded twice daily (Esselink, 2017). Soil pH (measured in $CaCl_2$) at the site ranges between 7.0 and 7.5 without pronounced differences between zones (Mueller et al. 2023). Both organic and inorganic carbon stocks increase from pioneer zone to high marsh (Mueller et al. 2023). $^{13}C$ signatures of the soil organic matter demonstrate a strong decrease in allochthonous marine-derived vs. autochthonous vascular plant-derived organic contributions along the flooding gradient from pioneer zone to high marsh (Mueller et al. 2023).

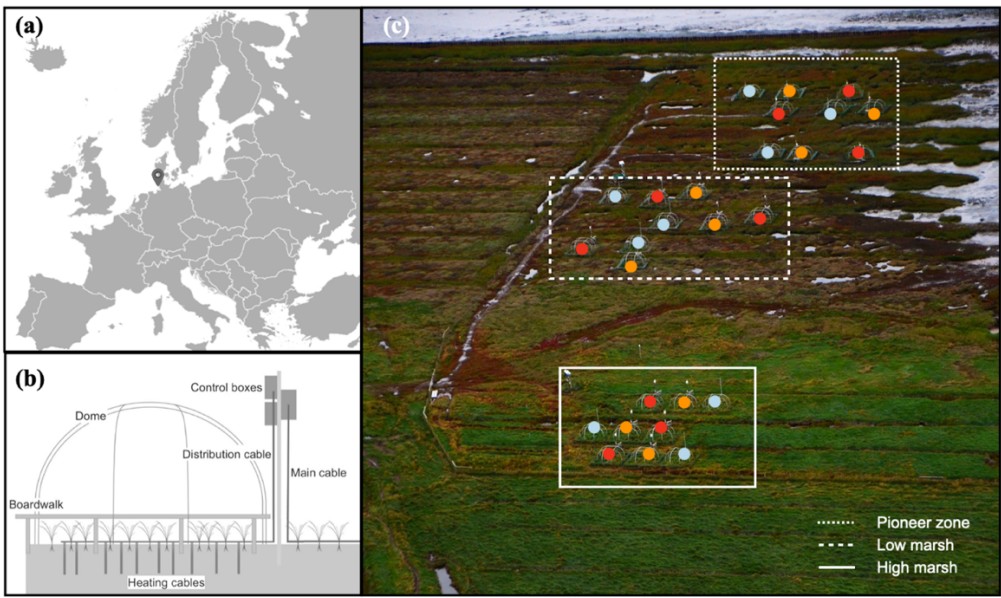

**Figure 1. Location of the MERIT experimental site (a), diagram of belowground and aboveground heating (b), and plot distribution in the pioneer zone, low marsh, and high marsh (c, aerial photo of the experimental site, courtesy of Norbert Kempf). Color points, i.e. blue, orange, and red, accordingly present the ambient, + 1.5 °C and + 3 °C treatments in different plots and marsh zones.**

## 2.2 Experimental design

MERIT consists of N = 27 experimental plots, each with an area of approx. 7 m². Belowground active temperature manipulation is conducted using 31 heating cables (GX 088L3100, 9.8 Ω/m, Danfoss, Denmark) per plot, inserted into the ground vertically to 1.0 m soil depth. Aboveground and soil-surface temperature manipulation is achieved using a combination of passive open-top chambers and surface heating cables with a length of 52 m per plot sinuously deployed at the soil surface (GX 088L3100, 9.8 Ω/m, Danfoss, Denmark) (Figure 1b). The experiment is conducted across three hydrological zones (pioneer zone, low marsh, and high marsh) and applies three temperature treatments (ambient, +1.5°C, and +3°C) in a full factorial design (3 zones x 3 temperature treatments x 3 replicates) (Figure 1d).

Belowground temperature was monitored continuously and logged at 5-min intervals using custom made thermistors and dataloggers. To control the heating rate evenly throughout the soil profile, sensors were placed at -5, -25, and -75 cm depth below the soil surface. At -5 cm, the highest variation in mean temperature difference across all marsh zones and plots ranged from 1.43° to 1.67° C for the +1.5° treatment, and 2.54° to 2.99° C for the +3.0° treatment. At -25 cm depth, the mean temperature difference values ranged from 1.51° to 1.55° C for the +1.5° treatment and 2.87° to 3.02° for the +3.0° treatment. At -75 cm, temperature difference values ranged from 1.14° to 1.43° C for the +1.5° treatment and 1.92° to 2.36° C for the +3.0° treatment (Rich et al. under review). We selected the belowground temperature data from -10 cm to -60 cm for the deployment times in 2018 and 2019 to calculate the average belowground ambient temperature per zone. Mean ambient temperature during the deployment time in 2018 (pioneer zone = 16.45 °C; low marsh = 15.99 °C; high marsh = 14.54 °C) was higher compared to 2019 (pioneer zone = 13.55 °C; low marsh = 12.95 °C; high marsh = 12.46 °C).

## 2.3 Decomposition of standardized plant litter

The initial decomposition rate ($k$) and stabilization factor ($S$) were assessed following the Tea Bag Index (TBI) protocol (Keuskamp et al. 2013). $k$ describes the rate at which the labile (here defined as hydrolysable) fraction of the plant material decomposes. $S$ describes the part of the labile fraction that did not decompose during deployment in the soil system and stabilized. Twelve polypropylene tea bags, six green tea (EAN: 8 714100 770603; Lipton, Unilever) and six rooibos tea bags (EAN: 8 711200 875665; Lipton, Unilever), were put into the salt marsh soil using two solid PVC-posts per plot, perforated with six holes between 10 and 60 cm soil depth (Figure 2). Posts contained either green or rooibos tea and were placed at a distance of 20 cm to each other (Figure 2). The initial weight of the tea bag content was determined by subtracting the mean content weight of 5 empty bags from the total tea bag weight (green tea: 1.592 ± 0.004 g; rooibos tea: 1.801 ± 0.006 g). Tea bags were deployed in two consecutive growing seasons. In 2018 (year 1 of the experiment), tea bags were deployed from 19 June to 20 September (= 93 days). In 2019 (year 2), tea bags were deployed from 14 May to 17 July (=93 days). Following deployment, tea bags were removed from the soil, and the tea material was carefully separated from roots and soil, dried for 48 h at 70 °C, and weighed. The calculation of $k$ and $S$ followed (Mueller et al. 2018), who used a different extraction protocol

than the original Keuskamp et al. (2013) publication, yielding slightly higher values for the hydroloyzable (= labile) fractions of green and rooibos tea.

(1) $W_r(t) = a_r e^{-kt} + (1-a_r),$

(2) $S = 1-a_g/H_g,$

(3) $a_r = H_r(1-S).$

$W_r(t)$ refer the weight of the rooibos substrate after the incubation time (t in days); $a_r$ refers the labile fraction substrate and $1-a_r$ refers the recalcitrant fraction of the rooibos substrate, respectively; $k$ is the initial decomposition rate; $S$ is the stabilization factor; $a_g$ represents the labile fraction of green tea substrate, and $H_g$ represents the hydrolysable fraction of the green tea substrate. The labile fraction of the rooibos substrate is calculated in Eq. (3) based on the hydrolysable fraction ($H_r$) and the stabilization factor $S$. $H_g$ and $H_r$ values are taken from Tang et al. (2021), because the same tea materials were used.

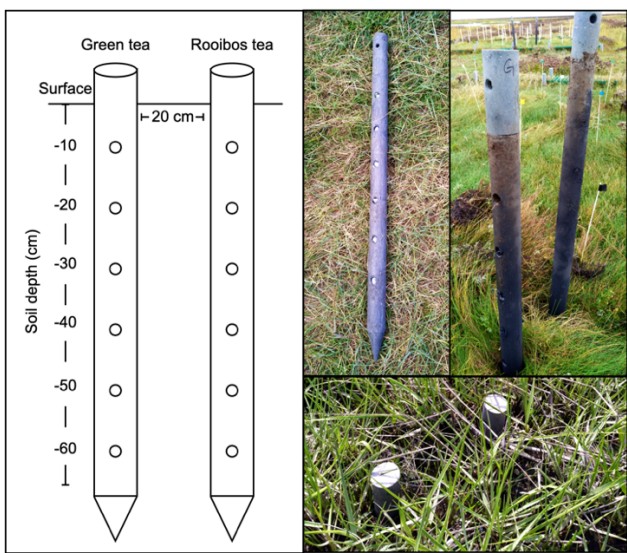

**Figure 2. Procedure for measuring standardized litter breakdown under in-situ warming. In each plot, two posts with six holes at a distance of 10 cm to -60 cm were put into the soil. One post contained six green tea bags, the other six rooibos tea bags.**

### 2.4 Characterization of soil redox conditions

A soil reduction index was determined based on the *Indicator of Reduction in Soils* technique (IRIS; *sensu* Jenkinson, 2002; Mueller et al., 2020; Rabenhorst, 2015). FeCl$_3$-coated PVC sticks were inserted to a soil depth of 30 cm in three zones along the marsh elevation gradient. There were 6 sticks per zone (n = 6), and 18 (N = 6 x 3) sticks per campaign, in total N = 72 (n = 4 campaigns x 18) sticks were analyzed. These measurements were conducted along a transect directly adjacent to the experimental plots. The Reduction Index describes the fraction of FeCl$_3$ paint that is removed from the PVC stick after four
150 weeks of deployment in the field. The IRIS method utilizes the property of the ferrihydrite paint to be reduced from solid-phase Fe(III) to soluble Fe(II) under anoxic soil conditions and in presence of microbial Fe-reducers. The area of removed paint from PVC sticks is used as a proxy for soil reduction (reduction Index). Upon the 4-wk deployment phase, sticks were removed from the soil and cleaned carefully with tap water to remove soil particles. Each stick was scanned to create digital images for further

processing. Image analysis was conducted applying a supervised classification using the software ArcGIS Pro. RGB color values (0-255) of 4300 randomly set points on a test set of sticks were determined. Points were classified as either reduced, not reduced or errors (background, scanning effects). The classification was included in a Random Forest model (confusion matrix 1.5%) using the software R. This model allowed for a pixel-wise classification of the scanned IRIS sticks. Sticks were analyzed in increments of 5 cm, covering a depth gradient from 0 to 30 cm. Reduction Index was calculated as an unitless value ranging from 0 to 1 based on the share of reduced pixels from the total pixels.

## 2.5 Statistical analyses

Two-way repeated-measures ANOVA were used to test the effects of warming treatment (ambient, +1.5°C and +3 °C), ecosystem zone (pioneer zone, low marsh, and high marsh), and soil depth (within subject / repeated measure) on TBI parameters for each year separately. Because we were not primarily interested in year-to-year differences, ANOVAs were used separately for each year to better understand the interactions between warming, zone, and soil depth. Pairwise comparisons were performed using Tukey's HSD tests. The normal distribution of residuals and homogeneity of variance were assessed visually and met ANOVA assumptions.

The temperature-induced change in $k$ and $S$ was calculated relative to the ambient controls ($k_0$) for each marsh zone separately: $\Delta k$ (%) = ($k_t/k_0$ - 1) x 100 and $\Delta S$ (%) = ($S_t/S_0$ - 1) x 100, t represents the value under different warming treatments (i.e. +1.5 and +3.0 ° C). Linear regression was used to analyze the relationships of $\Delta k$ and $\Delta S$ with soil depth.

These analyses were conducted using the statistical software STATISTICA, version 12 (StatSoft Inc, Tulsa, Oklahoma, USA).

## 3. Results

### 3.1 Decomposition and stabilization dynamics

Initial decomposition rate ($k$) was increased by warming in both year 1 and year 2 of the experiment (Figure 3 and Figure 4). The effect of the +3.0°C treatment (+158.6% in year 1; +234.6% in year 2) was more pronounced than that of the 1.5°C treatment (+162.3% in year 1; +170.39% in year 2). Overall, positive warming effects on $k$ appear to be consistent across marsh zones and soil depths (Figure 3a-c and Figure 4a-c), despite a significant depth x zone x warming interaction in year 2, as indicated by ANOVA ($p < 0.01$, Table 1). We ascribe this significant three-way interaction term to inconsistent depths trends of the warming responses between the marsh zones (Figure 4a-c). That is, the magnitude of the +3.0°C warming effect on $k$ decreases linearly with increasing soil depth in pioneer zone and high marsh, whereas the warming effect in the low marsh was maximized at an intermediate soil depth (Figure 4a-c).

The stabilization factor ($S$) was not affected by the warming treatment as a single factor, but by the depth x zone x warming interaction in year 2 of the experiment (Table 1). Negative effects of warming on $S$ were pronounced throughout the whole soil profile of the high marsh (on average -13.6% at +3.0°C; -8.6% at +1.5°C), but were restricted to the upper soil layers in the pioneer zone and low marsh (Figures 4d-f). The soil depth to which $S$ responded to warming treatments increased across the

marsh elevation gradient from pioneer zone via low to high marsh (Figure 4). While the magnitude of negative warming effects decreased with soil depth in both pioneer zone and low marsh, the opposite was true for the high marsh (Figure 5).

**Table 1 Results of two-way repeated-measures ANOVA testing for effects of warming (W), marsh zone (Z), soil depth (D), and their interactions on TBI parameters ($k$ = initial decomposition rate and $S$ = stabilization factor) in year 1 (2018) and year 2 (2019) of the experiment. Significant effects ($p \le 0.05$) are shown in bold.**

| | | Between subject | | | | Within-subject | | | | | | | | | |
|---|---|---|---|---|---|---|---|---|---|---|---|---|---|---|---|
| | | W | | Z | | W x Z | | D | | D x Z | | D x W | | D x Z x W | |
| | | *F* | *p* | *F* | *p* | *F* | *p* | *F* | *p* | *F* | *p* | *F* | *p* | *F* | *p* |
| *Year 1* | *k* | 10.67 | **0.00** | 3.05 | 0.08 | 0.89 | 0.50 | 10.00 | **0.00** | 0.48 | 0.90 | 1.02 | 0.44 | 0.39 | 0.99 |
| | *S* | 0.00 | 1.00 | 56.11 | **0.00** | 0.71 | 0.60 | 32.74 | **0.00** | 5.39 | **0.00** | 1.02 | 0.44 | 0.84 | 0.66 |
| *Year 2* | *k* | 36.32 | **0.00** | 0.17 | 0.85 | 0.58 | 0.68 | 10.92 | **0.00** | 1.77 | 0.10 | 1.76 | 0.10 | 2.23 | **0.01** |
| | *S* | 2.64 | 0.10 | 35.67 | **0.00** | 0.45 | 0.77 | 68.87 | **0.00** | 5.33 | **0.00** | 0.45 | 0.89 | 2.06 | **0.02** |

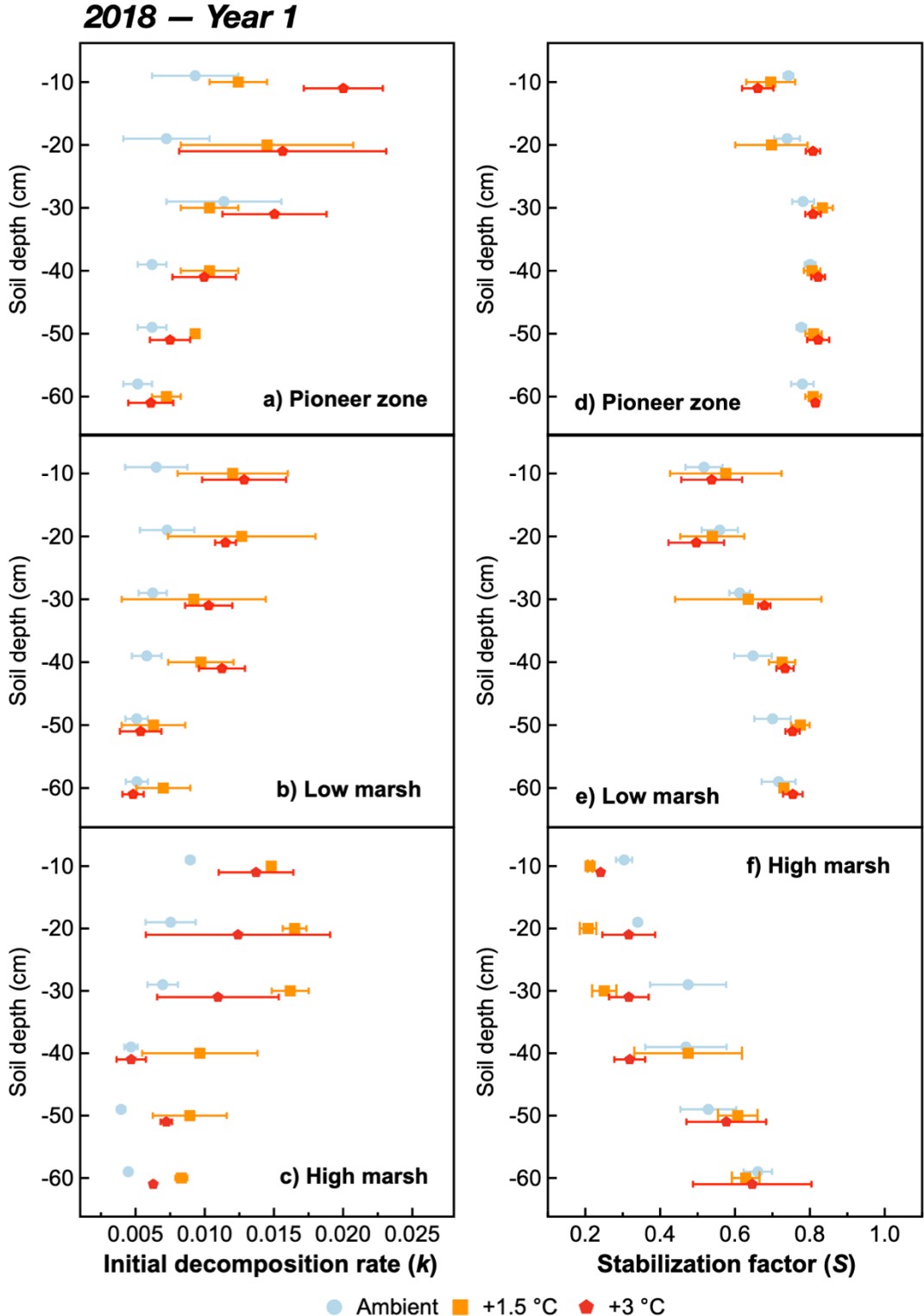

**Figure 3** The initial decomposition rate (*k*) (a, b, c) and stabilization factor (*S*) (d, e, f) at different soil depths of the pioneer zone, low marsh, and high marsh zones under three temperature treatments (ambient, + 1.5 °C and + 3 °C) in year 1 (2018). Values are means ± SE (n =3). *k* describes the labile fraction which is decomposed in the deployed material, and *S* presents the part of the labile fraction that did not decompose which stabilized in the soil.

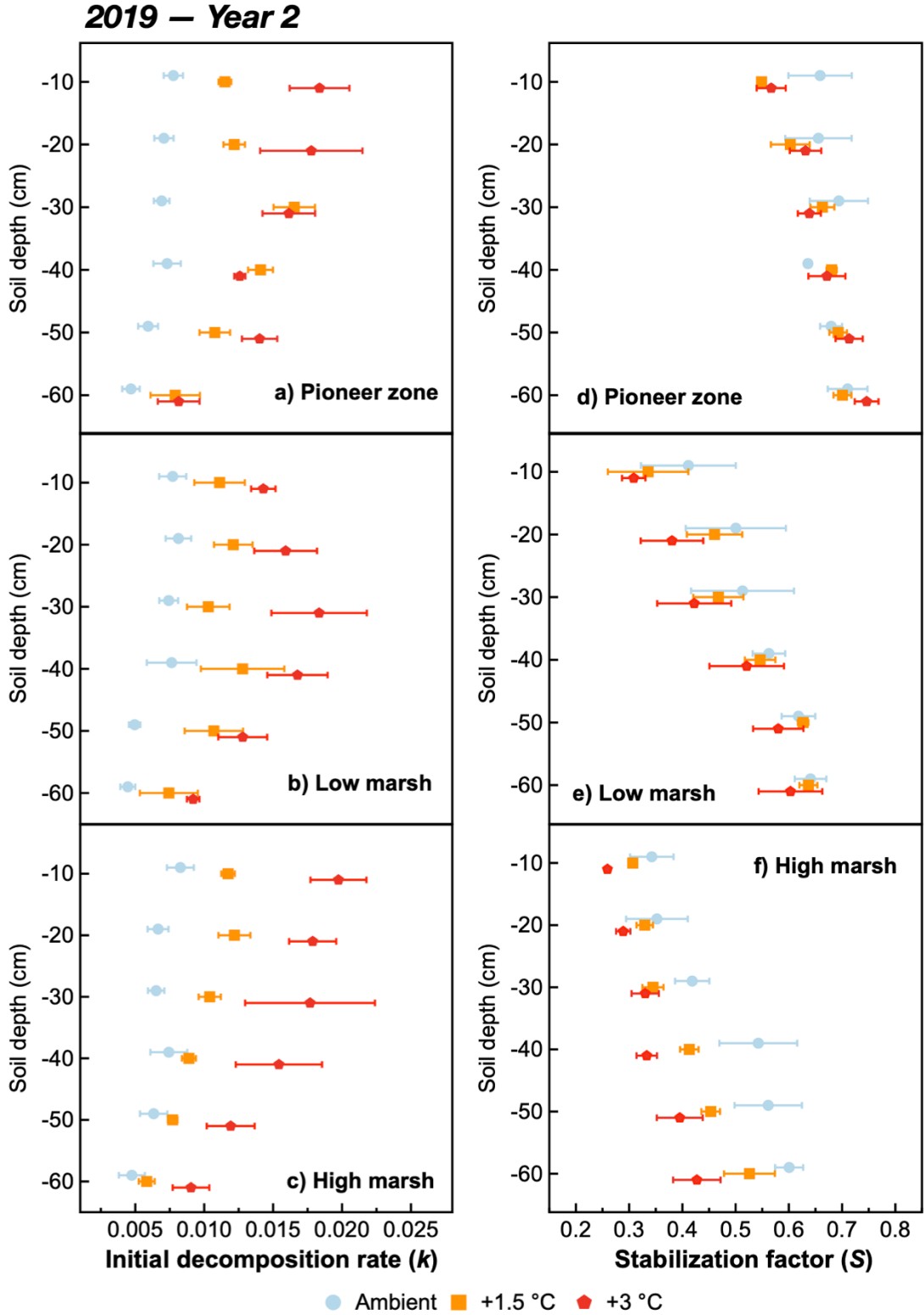

**Figure 4** The initial decomposition rate (a, b, c) and stabilization factor (d, e, f) at different soil depths of the pioneer zone, low marsh, and high marsh zones under three temperature treatments (ambient, + 1.5 °C and + 3 °C) in year 2 (2019). Values are means ± SE (n =3). *k* describes the labile fraction which is decomposed in the deployed material, and *S* presents the part of the labile fraction that did not decompose which stabilized in the soil.

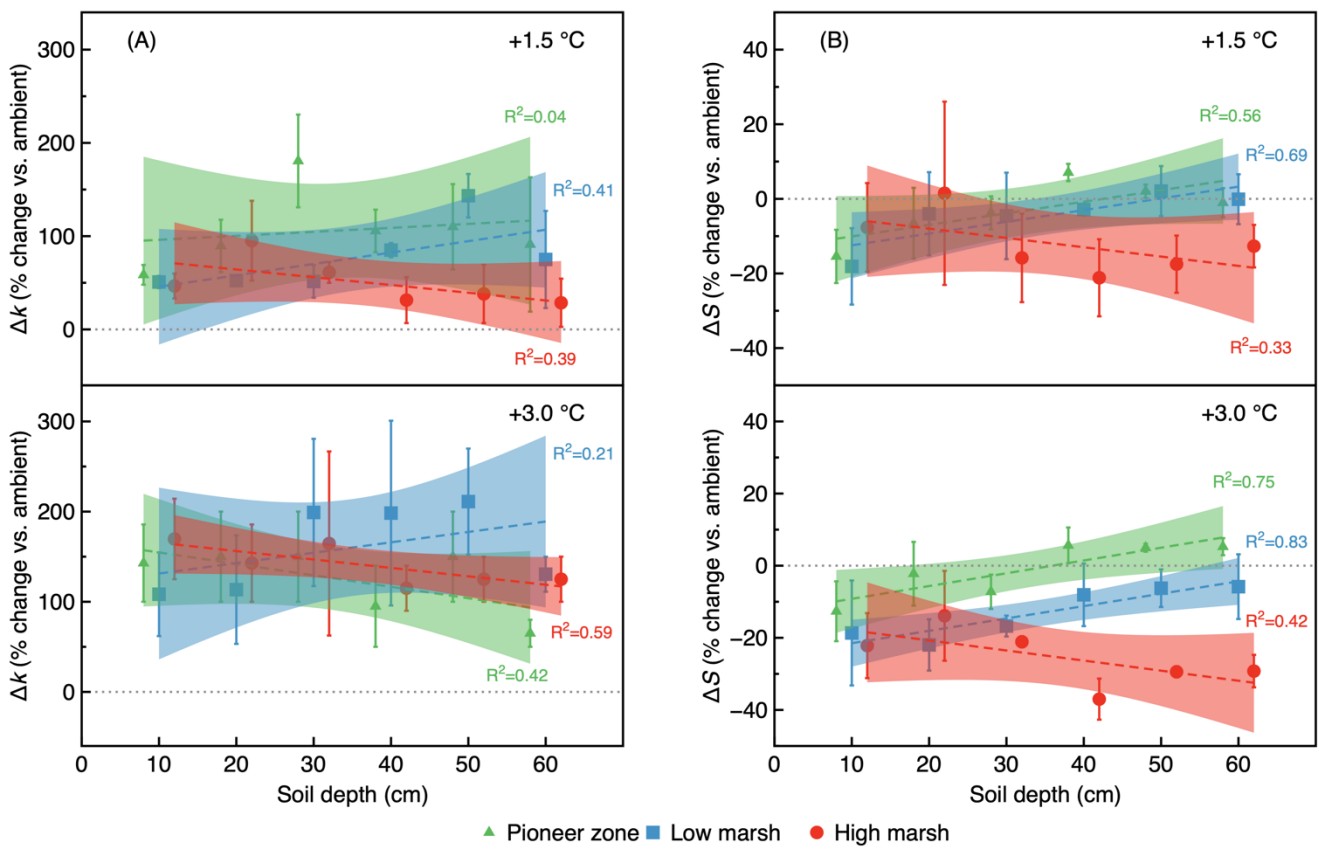

**Figure 5** Warming-treatment effect size (% change vs. ambient) as a function of soil depth and marsh zone, shown for the initial decomposition rate, *Δk* [A] and the stabilization factor, *ΔS* [B] in year 2 (2019). Linear regression was used to analyze the relationships of *Δk* and *ΔS* with soil depth.

## 3.2 Soil redox conditions

Pronounced soil redox gradients both with respect to soil depth and flooding frequency exist at the study site (Figure 6). Reducing soil conditions markedly increase with soil depth and along the flooding gradient from high marsh to pioneer zone. In the frequently flooded pioneer zone, reducing soil conditions are reached closer to the soil surface than in the high marsh (Figure 6).

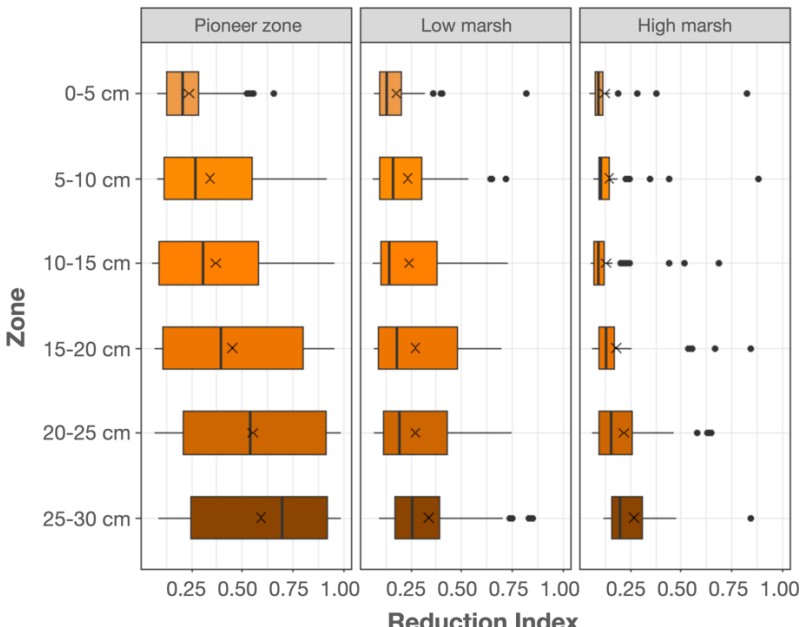

**Figure 6. Soil reduction in relation to marsh zone and soil depth at the Hamburger Hallig salt marsh site. Shown are median- (black bar) and mean values (black x). The box is giving the interquartile range, and potential outliers are depicted as black points. Data are based on n = 6 observations per zone, deployed over four consecutive deployment campaigns (July-October). The reduction index (0-1) describes the fraction of FeCl$_3$ paint that is removed from the PVC stick after four weeks of deployment in the field.**

## 4. Discussion

### 4.1 Litter breakdown parameters response to rising temperature

The initial decomposition rate ($k$) was strongly increased by warming across all marsh zones in two consecutive years (Table 1). By contrast, warming effects on stabilization ($S$) were less consistent and only present in year 2. This finding agrees with an large number of studies, from a range of ecosystems, demonstrating that $k$ and $S$ can be de-coupled, rather than strictly inversely related (Elumeeva et al., 2018; Fanin et al., 2020; Mori et al., 2022; Ochoa-Hueso et al., 2020; Sarneel et al., 2020; Sarneel and Veen, 2017; Tang et al., 2020).

Our results suggest that the decoupling of warming responses in $k$ and $S$ is controlled by hydrology or – more specifically – the soil redox status. Soil depth and marsh zone had no effects on $k$ (two-way ANOVA, $p > 0.1$), which shows that the initial decomposition rate of labile plant inputs is not influenced by hydrology or the soil redox status. This finding agrees with previous studies demonstrating the effects of oxygen availability on OM decomposition depends on OM quality, and that labile materials decompose at similar rates in oxic and anoxic environments (Benner et al., 1984; Kristensen et al., 1995). By contrast, $S$ increased with flooding frequency (i.e. from high marsh to pioneer zone) and with soil depth, indicating that the stabilization of labile materials does depend on the soil redox status. Flooding and soil depth were also the primary constraints of warming effects on $S$. Specifically, warming effects on $S$ were restricted to the topsoil in the pioneer zone, but the soil depth to which $S$ responded to warming treatments increased across the marsh elevation gradient via low to high marsh (Figure 4d-f).

The here observed redox constraints on warming effects resemble the findings of the SPRUCE (Spruce and Peatland Responses Under Changing Environments) warming experiment operating in a boreal peatland ecosystem in Minnesota, United States. Research in SPRUCE demonstrated a dramatic reduction of warming effects on soil OM decomposition in strongly reducing waterlogged subsoils compared to less reducing topsoils (Hopple et al., 2020; Wilson et al., 2016). It is possible that oxygen constraints on phenol-oxidase activity, following the enzymic-latch hypothesis (Freeman et al., 2001), are responsible for the redox control of warming effects on both the labile OM stabilization observed in our study and soil OM decomposition observed in SPRUCE. The enzymic-latch hypothesis states that phenolic substances accumulate under anoxic conditions, as phenol-oxidase activity requires oxygen, and inhibit the activity of hydrolases responsible for the breakdown of the majority of organic compounds supplied to the soil system. For blue carbon ecosystems, the phenolic-driven reduction of OM decomposition has recently been confirmed for the breakdown of sucrose in seagrass sediments (Sogin et al., 2022), demonstrating that even the breakdown of short-chained and labile sugars can be affected by the enzymic latch. The application of the enzymic latch hypothesis to the findings of the present study is not straightforward, because it is unclear how an accumulation of phenolics could increase the stabilization of labile OM, as well as the warming sensitivity of this process, but not their initial decomposition rate. It is however possible that initial microbial processing increased the secondary chemical recalcitrance of originally labile OM (Prescott 2010; Lützow et al. 2006) thereby increasing the sensitivity to phenolic inhibition of downstream enzymatic processes.

The current state of science lacks a mechanistic understanding of labile OM processing and stabilization in wetland soils, so that we cannot easily build hypotheses to explain the here observed redox control. For terrestrial soil systems, an increasing number of studies highlighted the importance of labile OM stabilization for long-term soil C storage (Frouz, 2018; Giannetta et al., 2022; Keiluweit et al., 2017; Lin et al., 2021; Lützow et al., 2006). Along with this, the prevailing concept of terrestrial soil OM formation was called into question stating that primarily recalcitrant OM inputs (i.e. non-hydrolysable compounds, such as phenolics) stabilize in the soil matrix and build the soil OM pool (e.g. Lützow et al., 2006; Schmidt et al., 2011). The Microbial Efficiency-Matrix Stabilization (MEMS) framework hypothesizes that labile plant inputs are the primary source for soil OM formation. Specifically, the MEMS framework considers labile plant inputs as the dominant microbial substrate source and thus, the dominant source of microbial decomposition products. Microbial decomposition products, in turn, are the main precursors of stabilized soil OM, which forms through aggregation or strong chemical bonding to the mineral matrix in terrestrial soils (Cotrufo et al., 2013). It is questionable if MEMS or related frameworks can be applied to wetland soils, because here physical protection through mineral armoring is largely absent in organic soils and hypothesized to be of little consequence in tidal mineral soils, which often lack aggregates owing to low fungal activity and wet–dry cycles (Kirwan and Megonigal, 2013; but see Spivak et al., 2019). We therefore hypothesize that secondary recalcitrance of originally labile organic compounds via microbial processing (Lützow et al., 2006) plays a larger role for the stabilization of labile OM inputs in many wetland soils than do aggregation and other interactions of OM with the mineral matrix.

Our data suggest that warming effects on litter breakdown are not necessarily linear. In year 1 of the experiment, 1.5°C warming had a stronger positive effect on the initial decomposition rate than 3.0°C warming in the high marsh. We doubt that potential

warming-induced drought effects in the soil surface were the responsible driver of this pattern, because higher $k$ under +1.5°C vs. +3.0°C warming was relatively consistent throughout the 60-cm soil profile. Instead, it is possible that plant-microbe interactions caused the observed pattern. For instance, warming-stimulated nutrient demands and belowground trait responses in plants could have negatively affected microbial communities via competition for nutrients (Noyce et al., 2019).

While qualitatively, warming effects on $k$ and $S$ were often similar between year 1 and 2, and effects were generally much more pronounced in year 2 compared to year 1 (Figures 3-4). The experimentally achieved temperature difference values were consistent between years (Table S1); however, differences in the actual temperature (not temperature difference) and the seasonal shift of 13 weeks between the two incubation periods of year 1 and 2 deployment phases could have affected the

280 magnitude of warming effects. Absolute soil temperatures were lower in year 2 than 1 (Figure S1), which could have resulted in the amplified warming effect. It is also possible that changes in the microbial community with increased treatment duration and/or greater microbial biomass as warming stimulated plant growth and substrate input to the soil system contributed to the observed effect amplification over time.

**4.2 Methodological considerations**

The use of PVC posts may have affected drainage and thus redox conditions of the deployed litter materials after flooding events. This could have amplified the redox differences between frequently and rarely flooded vegetation zones we observed. However, we argue that this potential effect on drainage is unimportant for the interpretation of our results, because our study was primarily designed to gain qualitative insight, not to capture actual rates of litter breakdown.

Compared to most previous studies which used the TBI to investigate litter breakdown in surface (approx. 5 cm depth) soils according to the original TBI protocol (e.g. Fanin et al., 2020; Marley et al., 2019; Mueller et al., 2018; Sarneel et al., 2020), our present study used the TBI to assess litter breakdown in whole-soil profiles (10 – 60 cm depth) in order to improve the mechanistic understanding of belowground carbon turnover. One important caveat in this respect is that we do not know how

TBI materials relate to the quality and microbial accessibility of native belowground inputs, namely root litter and rhizodeposits such as exudates. Warming and other climate change drivers are expected to induce changes in the quality of plant litter and other organic matter inputs accumulating in salt-marsh soils, for instance through shifts in the plant community composition that can potentially counterbalance or amplify the effects on decomposition processes described here (Mueller et al. 2018; 2020). Future research within the MERIT project will therefore address litter quality-feedback effects on decomposition

processes in order to gain a more complete understanding of warming effects on salt-marsh soil carbon cycling.

While the quality of TBI materials is likely to somewhat resemble the quality of root litter, it is questionable if this study can be used to infer anything about the turnover dynamics of root exudates – known to represent a considerable belowground carbon flux (Canarini et al., 2019). In this context, it is also important to note that the TBI, as well as native-litter bag techniques, likely

reduce the influence of aggregate protection in relation to plant inputs under natural conditions. It is therefore possible that important stabilization mechanisms of terrestrial soils, relying on aggregation or chemical interactions with the mineral matrix, cannot be adequately captured by the method. The importance of these stabilization mechanisms in wetlands soils, however, remains to be evaluated (Kirwan and Megonigal, 2013; Spivak et al., 2019).

A number of recent studies have highlighted the importance of dissolved organic carbon (DOC) and nutrient leaching during early litter breakdown in the context of the TBI (Gessner et al., 2010; Lind et al., 2022; Marley et al., 2019). Because leaching is a rapid process, particularly in wetlands, we assume that, in the present study, leaching was complete and thus, did not contribute to the observed variability in $k$ and $S$. Indeed, $k$ did not increase, and $S$ did not decrease with flooding (elevation gradient) or soil moisture (depth gradient) suggesting that leaching did not (overly) affect the variability of the data presented
here.

## 5 Conclusion

Our results show that warming can strongly increase the initial rate of labile litter decomposition, but has less consistent effects on the stabilization of this material. This finding suggests that warming may accelerate carbon and nutrient cycling through
stimulated initial decomposition rates, whereas soil organic matter formation and carbon sequestration through stabilization may be less consistently affected. We argue that the differential outcome of warming effects on initial decomposition rate and stabilization factor were mediated by the soil redox status, with redox conditions constraining the warming response of litter stabilization but not its initial decomposition rate. Because belowground organic matter turnover is a key determinant of surface elevation gain and carbon sequestration in blue carbon ecosystems, our findings may yield important implications for our
understanding of climate change effects on ecosystem stability and carbon sequestration in the coastal zone.

## Author contributions

HT, SN, KJ, and PM designed the TBI decomposition study.  SN, KJ, and RR designed the field experiment. HT conducted the TBI assays and analyzed the resulting data. JM conducted the study on soil reduction. HT and PM wrote the original draft
with input from all co-authors.

## Competing interests

The authors declare that they have no conflict of interest.

## Data availability

All data presented in this paper are available upon reasonable request.

**Financial support**

Hao Tang received financial support from the China Scholarship council (grant no. CSC201606910043). Peter Mueller was
340 supported by the DFG Emmy Noether program (grant no. 502681570), the DAAD (German academic exchange service) PRIME fellowship program funded through the German Federal Ministry of Education and Research (BMBF). Julian Mittmann-Goetsch received funding from the Fischer Stiftung (Stifterverband für die Deutsche Wissenschaft) as part of the Seal-C project (a Stability Assessment of Wadden Sea Blue Carbon Stocks).

**Acknowledgements**

Peter Mueller was funded by the Deutsche Forschungsgemeinschaft (DFG, German Research Foundation) within the Research Training Group 2530 (Biota-mediated effects on carbon cycling in estuaries: project number 407270017). We would like to thank Dr. Martin Stock, Armin Jess, and the administration of the Schleswig-Holstein Wadden Sea National Park for support and the opportunity to establish the experiment at their site. We would like to thank Detlef Böhm and the many colleagues of
350 the Applied Plant Ecology group who assisted with construction of the experiment, and especially Tom Kamin, Jörn Ehlers, and Stefan Knaak for their help and valuable technical expertise.

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
