# Peer review of "Warming accelerates belowground litter turnover in salt marshes – insights from a Tea Bag Index study"

_Biogeosciences, 2022_

## Author Comment (AC3)

We would like to thank three reviewers for their time and constructive comments. Below we respond to each comment separately (in blue font) referring to the line numbers of the original submission.

**Referee #1**

The paper represents a very important perspective of coastal ecosystem carbon dynamics which is not adequately understood. The findings and arguments of the study are critical and very important for future studies exploring coastal soil carbon dynamics which is susceptible to climate change. The manuscript is very well written, explained and discussed. Data are very well presented and clearly interpreted.

Experimental design and methods used to cover the whole soil profile is the significant development in the study. But the issue of whether TBI materials represent the real world scenario with regard to belowground biomass, litter and organic matter turnover remains critical.

We intend to elaborate on this point in the respective subsection of the Discussion, i.e. the Methodological considerations:

*The interpretation of results obtained from standardized approaches like the TBI needs to be made cautiously because litter quality is a key parameter controlling its decomposition and turnover dynamics. Because TBI materials differ from that of native salt-marsh plant litter, we did not expect to capture actual rates of litter breakdown and turnover with our approach. However, standardized approaches like the TBI, or the cotton-strip assay (e.g., Latter and Walton, 1988), are useful to separate the effects of abiotic factors, e.g. warming, from litter-quality effects on decomposition processes, thus facilitating an improved mechanistic understanding (Keuskamp et al. 2013; Tang et al. 2021; Mueller et al. 2018). Warming and other climate change drivers are expected to induce changes in the quality of plant litter and other organic matter inputs accumulating in salt-marsh soils, for instance through shifts in the plant community composition that can potentially counterbalance or amplify the effects on decomposition processes suggested here (Mueller et al. 2018; 2020). Future research within the MERIT project will therefore address litter-quality-feedback effects on decomposition processes in order to gain a more complete understanding of warming effects on salt-marsh soil carbon cycling.*

Whether the solid PVC posts and perforated holes in which tea bags were placed had any impact that could lead to different conditions in terms of soil moisture, temperature and microbial activity compared to natural soils around need to be discussed.

We will, likewise, elaborate on this point in the Methodological considerations section.

*The use of PVC posts may have affected drainage and thus redox conditions of the deployed litter materials after flooding events. This could have amplified the redox differences between frequently and rarely flooded vegetation zones we observed. We argue that this potential effect on drainage is unimportant for the interpretation of our results, because our study was designed to gain mechanistic insight, not to capture actual rates of litter breakdown (compare comment above).*

Besides mean elevation other edaphic characteristics of the three marsh types could be described under site description to signify the zonations.

Additional information on the variability in edaphic characteristics along the elevation gradient from pioneer zone to high marsh will be added to the site description. This will include soil salinity, pH, organic carbon content, inorganic carbon content, and del13C of the accumulated organic matter as a proxy of organic matter origin.

**Referee #2**

The manuscript presents a study of belowground decomposition processes in salt marshes in response to climate warming, which are underexplored. The experiment has a novel experimental design that makes use of two warming treatments along a flooding gradient, where decomposition rates and stabilization factor were assessed throughout the soil profile using a standardized litter bag method (tea bag index;TBI). In addition, soil redox index was measured along this gradient to deduce whether changes in hydrology and redox conditions affect decomposition.

The study design has multiple aspects and gives interesting new insights to decomposition processes in this system and the manuscript is in general well written, though there are some sections that need further clarification. I also have more substantial comments regarding the methodology and performed analyses that I will detail below.

Since the experiment was performed in salt marshes, leaching could play a large role in mass loss due to high soil moisture/inundation and might influence the findings, see for example (Gessner et al. 2010, Lind et al. 2022, Marley et al. 2019). While the authors do state that they used a tidal wetland-adapted TBI protocol, I would like to see details on what adjustments this protocol has for the k and S calculations within the manuscript and whether this takes into account leaching. In addition, it might be good to raise and discuss this point already in the introduction.

We will provide more detail on the tidal wetland-adapted protocol (sensu Mueller et al. 2018, biogeosciences) in the Methods. Mueller and colleagues protocol assumes a higher hydrolyzable fraction than that of Keuskamp and colleagues (2013). It does not explicitly consider leaching. It is important to note that k did not increase, and S did not decrease with flooding (elevation gradient) or soil moisture (depth gradient) suggesting that leaching did not (overly) control our results.

We are aware of the above-mentioned studies highlighting the potential relevance of leaching and will cite them in the revised version. Because leaching is a rapid process, particularly in wetlands, we assume that leaching was complete throughout all vegetation zones and thus, did not contribute to the observed variability in k and S. We will elaborate on this point in the Discussion section (i.e. Methodological considerations).

I also wondered whether the use of PVC tubes could influence the conditions in which the decomposition experiment was performed, as the solid pipes might prevent water flow and could potentially also hinder the warming treatment used. Could the authors address this point. Potentially assess whether the temperature treatment was affected by the use of solid PVC tubes?

We will elaborate on this point in the respective subsection of the Discussion (i.e. Methodological considerations).

*The use of PVC posts may have affected drainage and thus redox conditions of the deployed litter materials after flooding events. This could have amplified the redox differences between frequently and rarely flooded vegetation zones we observed. We argue that this potential effect on drainage is unimportant for the interpretation of our results, because our study was primarily designed to gain mechanistic insight, not to capture actual rates of litter breakdown.*

*Concerning the temperature effect: The temperature of the PVC posts will equilibrate to the surrounding soil mass of the 7-m2 plot (= 7 m3 of heated soil), which far exceeds the thermal mass of the posts. PVC is a poor thermal conductor and would not be able to conduct heat vertically in soils.*

This also leads me to another point that I would like clarification on being the temperature monitoring. The authors only provide a mean temperature during the deployment time for the different zones. Which leaves me to wonder whether temperature was also monitored in the various treatments, to check whether treatments were effective. I could also not find at which depth temperature was monitored, only 1 depth or throughout soil profile? Could the authors provide a figure with the temperatures for the different zones and treatments throughout the deployment times to give a better representation of the treatments used?

Belowground temperatures were monitored continuously and logged at 5-min intervals. To control the heating rate evenly throughout the soil profile, sensors were placed at 5, 25, and 75 cm depth. At 5 cm, the highest variation in mean delta temperature across all marsh zones and plots ranged from 1.43° to 1.67° C for the +1.5° treatment, and 2.54° to 2.99° C for the +3.0° treatment. At 25 cm depth, we observed delta values ranging from 1.51° to 1.55° C for the +1.5° treatment and 2.87° to 3.02° for the +3.0° treatment. At 75 cm, delta values ranged from 1.14° to 1.43° C for the +1.5° treatment and 1.92° to 2.36° C for the +3.0° treatment (Rich et al. under review, Ecosystems).

A methods paper providing these and more details is currently under review. Therefore, we will not add result figures on the effectiveness of warming treatments to the present ms. Instead, we will add a brief summary on this to the Methods section and refer the reader to methods paper (Rich et al. under review). We will also provide a figure showing mean temperatures and mean delta temperatures measured during the 2019 season. This figure can be shared confidently with the reviewers but not beyond.

The description of the statistical analyses is different from the results you present. From the results I deduce that you have performed separate analyses of warming and zone and their interaction for the different years (between subject), excluding the effect of depth.

Then there is separate analysis of the effect of depth, depth*zone and depth*zone*warming (within subject). In the statistical analyses section these are lumped together and it is not clear that the analyses were performed for each year separately. Furthermore, it is not clear to me why these analyses are split and what the authors mean with the indication "between subject" and "within-subject"? The "within subject" analysis would still need warming and zone included as separate factors to account for their effect.

We used the two-way repeated-measures ANOVA to test the effects of warming, zone, and depth (**depth =within subject / repeated measure**) for each year separately.

We were not primarily interested in year-to-year differences and therefore used ANOVAs separately for each year to better understand the interactions between warming, zone, and soil depth.

We will improve clarity of the statistical methods section in the revised ms.

Furthermore, there is no mention of analysis done to produce figure 5 and it is hardly discussed in the results. Lacking this information it is hard to properly assess the results.

The temperature-induced change in k and S (deltak)was calculated relative to the ambient controls (k0) for each marsh zone separately: $\Delta k$ (%) = $(k_t/k_0 - 1)$ x 100 and $\Delta S$ (%) = $(S_t/S_0 - 1)$ x 100, t represents the value under different warming treatments (i.e. +1.5 and +3.0 ° C). Linear regression was used to analyze the relationships of $\Delta k$ and $\Delta S$ with soil depth. This information will be added to the Methods as well as to the caption of figure 5 in the revised ms.

Personally, I think Figure 5 gives a much better look into how the different factors influence decomposition rate k and S. I think expanding on this analysis would improve the manuscript as it sheds more light what factors/conditions affect decomposition in salt marshes. This could also be a way Figure 3 and 4 could then potentially be moved to supplementary material.

We agree with the reviewer that there is some redundancy between the figures and that Figure 5 yields more information than Figures 3-4. We are happy with the idea of moving Figures 3-4 to the Supplement, in case the editor agrees on this.

I also wondered why the authors did not use the measured soil reduction index in their analysis, as they as they discuss the influence of redox a lot in the discussion, but have not directly tested these links in their analysis. Why not use reduction index as predictor of k and S?

We did not quantify soil reduction inside the experimental plots but along a transect directly adjacent to the plots. Thus, there are no data on warming effects on soil reduction. Soil reduction was assessed to characterize the three marsh zones across which the experimental plots are distributed, i.e., pioneer zone, low marsh, high marsh. We originally intended to present this information in the Methods section just like we provided other aspects of the site and zone characterization here (e.g. plant-community composition, surface elevation) but were asked by the editor to move it to the results section prior to peer review. In any case, we believe these are important data to show that soil redox conditions actually differ between marsh zones and with respect to soil depth. However, we cannot use these data for e.g. correlations with S or k.

Specific comments

line 68. What do the authors mean with short- and mid-term warming?

Short- and mid-term was intended to refer to year 1 and 2 of the experiment, respectively. However we are not using this concept in the later parts of the ms and will therefore remove this and related statements.

Line 96 Can the authors also indicate the location of the different warming treatments. And were these treatments randomly assigned?

Treatments were randomly assigned. Locations will be added to the figure.

Line 132. Not clear if it is 1 PVC stick per zone or whether there is replication? The transect in the next line adds to my confusion. Figure 6 legend states "n = 6 observations per zone, deployed over four consecutive deployment campaigns (July-October)". Clarify in methods.

There were 6 sticks per zone (n = 6); in total 3 x 6 = 18 sticks per campaign. There were 4 campaigns. So in total 4 x 18 = 72 sticks were analyzed. We will improve the description for the revised version.

Figure 3 Hard to read this figure as the error bars of the different treatments are overlapping. Please adjust figure so it is possible to discern the different treatments per soil depth.

Figure layout will be adjusted or moved to Supplement.

Line 114 Incubation period different in 2018 and 2019, June-Sept vs May-July. Why? This does explain why temperatures were higher in 2018 as it is later in the season.

We agree with the reviewer and will add this aspect to the Discussion section. The two incubation experiments were started after the warming treatment was switched on in each year:  In year 1, warming was switched on in May, in year 2, warming started in April. We were not primarily interested in year-to-year differences, but repeated the incubation in order to test if results from year 1 are replicable. We will make this clearer in the revised version.

Line 150-153 It is not clear to me what the authors are trying to say. Clarify

Will be rephrased.

Figure 3c Why high marsh higher k in +1.5 vs +3.0 treatment?

This is indeed an interesting pattern. If higher k at 1.5 vs 3.0 was only observed at the soil surface, drought effects could have played a role in the less-frequently flooded high marsh. However, the pattern is relatively consistent throughout the soil profile. We will elaborate on this point in the revised Discussion section, focusing on plant-microbe interactions and nutrient constraints as

potential drivers of the effect (compare Noyce et al. 2019). It is also important to stress that the pattern disappears in year 2.

Why k higher in warming treatments in 2019 vs 2018? Bigger difference in temperature?

Temperature differences were consistent between years: The mean delta temperature across all marsh zones and plots ranged from 0.99° to 1.66° C for the +1.5°C treatment, and 1.61° to 2.81° C for the +3.0°C treatment in 2018, and the mean delta temperature across all marsh zones and plots ranged in 2019 from 1.26° to 1.82° C for the +1.5°C treatment, and 1.69° to 2.81° C for the +3.0°C treatment (Table 1). One potential reason is greater microbial biomass in year 2 vs. 1 of the experiment as warming stimulated plant growth and substrate input to the soil system. However, also differences in the actual temperature (not delta T) and the slight seasonal shift between the two incubation periods could have affected the magnitude of warming effects. We will discuss this point in the revised Discussion.

Table 1 Average belowground temperature across marsh zones and warming treatments in 2018 and 2019. Value are means ± SE (n =3).

| Year | Treatment | Pioneer zone | Low marsh | High marsh |
|------|-----------|--------------|-----------|------------|
| 2018 | Ambient | 16.45 ± 0.12 | 15.99 ± 0.14 | 14.54 ± 0.44 |
| | + 1.5°C | 17.66 ± 0.11 | 16.98 ± 0.29 | 16.20 ± 0.22 |
| | + 3°C | 18.06 ± 0.67 | 18.07 ± 0.37 | 17.35 ± 0.09 |
| 2019 | Ambient | 13.55 ± 0.33 | 12.95 ± 0.10 | 12.46 ± 0.03 |
| | + 1.5°C | 15.37 ± 0.14 | 14.21 ± 0.22 | 13.89 ± 0.04 |
| | + 3°C | 16.36 ± 0.11 | 15.01 ± 0.14 | 14.15 ± 0.42 |

Technical corrections:

Line 49 "in" question

Will be changed accordingly.

Line 50. Do the authors mean labor intensive instead of efficient? Constructing a lot of litter bags is labor intensive in my opinion.

Will be improved accordingly.

Line 54 rephrase: represents a widely used standardized litter bag approach

Will be changed accordingly.

Line 68 replace "over" with "for". Over implies the incubation time was either 1 or 2 years, but there were separate 3-month incubations in each year.

Will be changed accordingly.

Line 81 Can the authors use a more widely known standard like meter above sea level for to indicate elevation instead of NHN?

Will be changed accordingly.

Line 132 "." After citation

Will be changed accordingly.

Line 132 remove "from pioneer…. High marsh."

Will be changed accordingly.

Line 248 known

Will be changed accordingly.

**Referee #3**

This paper examines patterns of initial decomposition rate and stabilization factor over a flooding gradient and depth profile in a tidal marsh. The focus on decomposition over an elevation gradient and belowground, where the greatest contribution by plants to blue C accumulation occurs, represents an important contribution to the literature. Further, the use of standard substrate to control for litter quality allows for a focus on abiotic drivers of decay. Overall, this is a good paper that, with some clarifications to the methods and stats and additional interpretation of results, will add valuable insights to decomposition processes in tidal marshes that are especially vulnerable to climate change.

Specific comments and questions:

L60-61: clarify by explaining what their proposed mechanism is for how the lack of oxygen inhibits warming effects.

We will provide more background here, focusing on the enzymic latch hypothesis and the recalcitrance of chemically stable tissues, such as phenolic compounds, in the absence of oxygen.

L63: reiterate that the use of a standard substrate is needed to understand warming effects.

Will be changed accordingly.

L70: is the first hypothesis expected regardless of soil depth?

Yes, the first hypothesis is meant to be general, and the second hypothesis is meant to further specify and explore the interaction effects with marsh zone and soil depth.

Site description – Figure 1 is difficult to see, and the zones are not clearly defined in the text. It would be helpful to explain how the zones are oriented relative to the open water, with the pioneer zone along the shoreline and the high marsh farthest inland. Also, how is "pioneer zone" defined?

The figure 1 will be improved accordingly, and more detail will be added to the site description.

*The pioneer zone is a typical feature of NW European salt marshes (and elsewhere) and is typically distinguished from the low marsh in studies from this region. The pioneer zone is defined as the area where pioneer vegetation covers ≥5 % (Petersen et al., 2014). In the Wadden Sea region its average surface elevation is below mean high tide. Thus, the pioneer zone is typically flooded twice daily (Esselink et al. 2017). This information will be added to the revised ms.*

Experimental design – How was the soil warming established and verified along the soil depth gradient? Was there uniform warming of the soil column or did it vary with depth? It would be nice to see a graph of these data. Did you confirm treatment conditions of +1.5 and +3 degrees warming? Why was the average soil temperature from -10 and -60 cm used as opposed to looking at temperature along the soil depth gradient at the same intervals at which decomposition and soil reduction were measured?

Belowground temperatures were monitored continuously and logged at 5-min intervals. To control the heating rate evenly throughout the soil profile, sensors were placed at 5, 25, and 75 cm depth. At 5 cm, the highest variation in mean delta temperature across all marsh zones and plots ranged from 1.43° to 1.67° C for the +1.5° treatment, and 2.54° to 2.99° C for the +3.0° treatment. At 25 cm depth, we observed delta values ranging from 1.51° to 1.55° C for the +1.5° treatment and 2.87° to 3.02° for the +3.0° treatment. At 75 cm, delta values ranged from 1.14° to 1.43° C for the +1.5° treatment and 1.92° to 2.36° C for the +3.0° treatment (Rich et al. under review, Ecosystems).

More detail is provided to a similar comment by Reviewer #1.

Decomposition – why was this examined across two different periods (June-Sept vs. May-July) in the two years? It is not surprising that ambient temperatures were cooler in year 2 (late spring/early summer) than in year 1 (late summer), which may have contributed to the larger effect sizes of

warming in year 2 compared to year 1. Address why these time periods were selected, and later discuss how this could have affected results.

We agree with the reviewer and will add this aspect to the Discussion section. The two incubation experiments were started after the warming treatment was switched on in each year: In year 1, warming was switched on in May, in year 2, warming started in April. We were not primarily interested in year-to-year differences, but repeated the incubation in order to test if results from year 1 are replicable. We will make this clearer in the revised version.

Statistics – This section is lacking details and does not fully track with the results presented. Were years compared statistically or tested separately? Why or why not? How were the effect sizes determined and analyzed, and why was this only examined in year 2? How was soil reduction analyzed?

We used the two-way repeated-measures ANOVA to test the effects of warming, zone, and depth (depth =within subject / repeated measure) for each year separately.

We were not primarily interested in year-to-year differences and therefore used ANOVAs separately for each year to better understand the interactions between warming, zone, and soil depth.

Based on the results of two-way repeated-measure ANOVA, the significant warming x depth x zone interaction was detected in year 2 only. A more detailed presentation of effect sizes was therefore restricted to year 2.

We will improve clarity of the statistical methods section in the revised ms.

Soil reduction: this parameter was assessed using the Indicator of Reduction in Soils technique. We agree that the information given on the methods is insufficient. We will provide a thorough description in the revised Methods section.

For the discussion and methodological considerations, how much could leaching be contributing to the results and different findings for k and S along the flooding gradient? How did the PVC influence the hydrology or connectivity of the tea bags with their surroundings? Was the temperature gradient verified within those PVC pipes? It would also be useful to revisit the importance of litter quality, as well as species-specific differences in decay with species turnover along the elevation gradient. While this study was designed to avoid plant influences, brief discussion of how it could affect these patterns, and how shifts in community composition with sea-level rise is another climate change driver to be considered that, if species differ in their contributions to blue C, could have implications for marsh resilience.

Similar comments with respect to leaching, PVC-post effects on the abiotic environment, and litter quality were made by R1 and R2 (compare above). The section on Methodological considerations will be improved in accordance.

Technical comments:

L13: clarify "plant production"

Will be changed accordingly.

L15: suggest "entire intertidal flooding gradient"

Will be changed accordingly.

L17: delete "of" before "(k)"

Will be changed accordingly.

L54: delete "probably"

Will be changed accordingly.

L59: offset "and thus strongly reducing" with commas

Will be changed accordingly.

L69: what is short- and mid-term warming effects? Is this in reference to projected warming of +1.5 vs. 3 degrees?

Will be changed accordingly.

L71: combine sentences so that it reads "…soil, and (2) that warming…"

Will be changed accordingly.

L77: "has operated" instead of "operates"

Will be changed accordingly.

L79: change comma after climate to semicolon

Will be changed accordingly.

L148: should this be "appear to be consistent"?

Will be changed accordingly.

L150: this is unclear. What do you mean by "refer the significant interaction"?

Will be changed accordingly.

L152: clarify that the relationship is "with increasing soil depth"

Will be changed accordingly.

L192: change to "a large" instead of "an"

Will be changed accordingly.

L209: add a comma after the citation

Will be changed accordingly.

L248: "known"

Will be changed accordingly.

---

## Author Response (AR1)

We would like to thank three reviewers for their time and constructive comments. Below we respond to each comment separately (in blue font) referring to the line numbers of the original submission.

**Referee #1**

The paper represents a very important perspective of coastal ecosystem carbon dynamics which is not adequately understood. The findings and arguments of the study are critical and very important for future studies exploring coastal soil carbon dynamics which is susceptible to climate change. The manuscript is very well written, explained and discussed. Data are very well presented and clearly interpreted.

Experimental design and methods used to cover the whole soil profile is the significant development in the study. But the issue of whether TBI materials represent the real world scenario with regard to belowground biomass, litter and organic matter turnover remains critical.

The interpretation of results obtained from standardized approaches like the TBI needs to be made cautiously because litter quality is a key parameter controlling its decomposition and turnover dynamics. Because TBI materials differ from that of native salt-marsh plant litter, we did not expect to capture actual rates of litter breakdown and turnover with our approach. However, standardized approaches like the TBI, or the cotton-strip assay (e.g., Latter and Walton, 1988), are useful to separate the effects of abiotic factors, e.g. warming, from litter-quality effects on decomposition processes, thus facilitating an improved mechanistic understanding (Keuskamp et al. 2013; Tang et al. 2021; Mueller et al. 2018). We elaborated on this point in the respective subsection of the Discussion, i.e. the Methodological considerations (line 300-304):

*"Warming and other climate change drivers are expected to induce changes in the quality of plant litter and other organic matter inputs accumulating in salt-marsh soils, for instance through shifts in the plant community composition that can potentially counterbalance or amplify the effects on decomposition processes described here (Mueller et al. 2018; 2020). Future research within the MERIT project will therefore address litter quality-feedback effects on decomposition processes in order to gain a more complete understanding of warming effects on salt-marsh soil carbon cycling."*

Whether the solid PVC posts and perforated holes in which tea bags were placed had any impact that could lead to different conditions in terms of soil moisture, temperature and microbial activity compared to natural soils around need to be discussed.

We elaborated on this point in the Methodological considerations section (line 290-293).

*The use of PVC posts may have affected drainage and thus redox conditions of the deployed litter materials after flooding events. This could have amplified the redox differences between frequently and rarely flooded vegetation zones we observed. However, we argue that this potential effect on drainage is not critical unimportant for the interpretation of our results, because our study was primarily designed to gain mechanistic insight, not to capture actual rates of litter breakdown. (compare comment above).*

Besides mean elevation other edaphic characteristics of the three marsh types could be described under site description to signify the zonations.

*Additional information on the variability in edaphic characteristics along the elevation gradient from pioneer zone to high marsh have been added to the site description (line 87-91).*

*"Soil pH (measured in $CaCl_2$) at the site ranges between 7.0 and 7.5 without pronounced differences between zones (Mueller et al. 2023). Both organic and inorganic carbon stocks increase from pioneer zone to high marsh (Mueller et al. 2023). $Del^{13}C$ signatures of the soil organic matter demonstrate a strong decrease in allochthonous marine-derived vs. autochthonous vascular plant-derived organic contributions along the flooding gradient from pioneer zone to high marsh (Mueller et al. 2023)."*

**Referee #2**

The manuscript presents a study of belowground decomposition processes in salt marshes in response to climate warming, which are underexplored. The experiment has a novel experimental design that makes use of two warming treatments along a flooding gradient, where decomposition rates and stabilization factor were assessed throughout the soil profile using a standardized litter bag method (tea bag index;TBI). In addition, soil redox index was measured along this gradient to deduce whether changes in hydrology and redox conditions affect decomposition.

The study design has multiple aspects and gives interesting new insights to decomposition processes in this system and the manuscript is in general well written, though there are some sections that need further clarification. I also have more substantial comments regarding the methodology and performed analyses that I will detail below.

Since the experiment was performed in salt marshes, leaching could play a large role in mass loss due to high soil moisture/inundation and might influence the findings, see for example (Gessner et al. 2010, Lind et al. 2022, Marley et al. 2019). While the authors do state that they used a tidal wetland-adapted TBI protocol, I would like to see details on what adjustments this protocol has for the k and S calculations within the manuscript and whether this takes into account leaching. In addition, it might be good to raise and discuss this point already in the introduction.

*We provided more detail on the tidal wetland-adapted protocol (sensu Mueller et al. 2018, biogeosciences) in the Methods (line 129-131). Mueller and colleagues protocol assumes a higher hydrolyzable fraction than that of Keuskamp and colleagues (2013). It does not explicitly consider leaching. It is important to note that k did not increase, and S did not decrease with flooding (elevation gradient) or soil moisture (depth gradient) suggesting that leaching did not (overly) control our results.*

*We are aware of the above-mentioned studies highlighting the potential relevance of leaching and have cited them in the revised version. Because leaching is a rapid process, particularly in wetlands, we assume that leaching was complete throughout all vegetation zones and thus, did not contribute to the observed variability in k and S. We elaborated on this point in the Methodological considerations (line 314-318).*

*"A number of recent studies have highlighted the importance of leaching in the context of the TBI (Gessner et al., 2010; Lind et al., 2022; Marley et al., 2019). Because leaching is a rapid process, particularly in wetlands, we assume that leaching was complete throughout all vegetation zones and*

*thus, did not contribute to the observed variability in k and S. Indeed, k did not increase, and S did not decrease with flooding (elevation gradient) or soil moisture (depth gradient) suggesting that leaching did not (overly) control our results."*

I also wondered whether the use of PVC tubes could influence the conditions in which the decomposition experiment was performed, as the solid pipes might prevent water flow and could potentially also hinder the warming treatment used. Could the authors address this point. Potentially assess whether the temperature treatment was affected by the use of solid PVC tubes?

We elaborated on this point in the respective subsection of the Methodological considerations (line 290-293).

*"The use of PVC posts may have affected drainage and thus redox conditions of the deployed litter materials after flooding events. This could have amplified the redox differences between frequently and rarely flooded vegetation zones we observed. However, we argue that this potential effect on drainage is not critical unimportant for the interpretation of our results, because our study was primarily designed to gain mechanistic insight, not to capture actual rates of litter breakdown."*

*Concerning the temperature effect: The temperature of the PVC posts will equilibrate to the surrounding soil mass of the 7-m2 plot (= 7 m3 of heated soil), which far exceeds the thermal mass of the posts. PVC is a poor thermal conductor and would not be able to conduct heat vertically in soils.*

This also leads me to another point that I would like clarification on being the temperature monitoring. The authors only provide a mean temperature during the deployment time for the different zones. Which leaves me to wonder whether temperature was also monitored in the various treatments, to check whether treatments were effective. I could also not find at which depth temperature was monitored, only 1 depth or throughout soil profile? Could the authors provide a figure with the temperatures for the different zones and treatments throughout the deployment times to give a better representation of the treatments used?

Belowground temperature was monitored continuously and logged at 5-min intervals using custom made thermistors and dataloggers. To control the heating rate evenly throughout the soil profile, sensors were placed at -5, -25, and -75 cm depth below the soil surface. At -5 cm, the highest variation in mean delta temperature across all marsh zones and plots ranged from 1.43° to 1.67° C for the +1.5° treatment, and 2.54° to 2.99° C for the +3.0° treatment. At -25 cm depth, the mean delta values ranged from 1.51° to 1.55° C for the +1.5° treatment and 2.87° to 3.02° for the +3.0° treatment. At -75 cm, delta values ranged from 1.14° to 1.43° C for the +1.5° treatment and 1.92° to 2.36° for the +3.0° treatment (Rich et al. under review). This information has been added in the Method part (line 106-111).

A methods paper providing these and more details is currently under review. Therefore, we will not add result figures on the effectiveness of warming treatments to the present ms. Instead, we will add a brief summary on this to the Methods section and refer the reader to methods paper (Rich et al. under review). We will also provide a figure showing mean temperatures and mean delta

temperatures measured during the 2019 season. This figure can be shared confidently with the reviewers but not beyond.

The description of the statistical analyses is different from the results you present. From the results I deduce that you have performed separate analyses of warming and zone and their interaction for the different years (between subject), excluding the effect of depth.

Then there is separate analysis of the effect of depth, depth*zone and depth*zone*warming (within subject). In the statistical analyses section these are lumped together and it is not clear that the analyses were performed for each year separately. Furthermore, it is not clear to me why these analyses are split and what the authors mean with the indication "between subject" and "within-subject"? The "within subject" analysis would still need warming and zone included as separate factors to account for their effect.

We used the two-way repeated-measures ANOVA to test the effects of warming, zone, and depth (**depth =within subject / repeated measure**) for each year separately.

We were not primarily interested in year-to-year differences and therefore used ANOVAs separately for each year to better understand the interactions between warming, zone, and soil depth.

We improved clarity of the statistical methods section in the revised ms (line 163-166).

*"Two-way repeated-measures ANOVA were used to test the effects of warming treatment (ambient, +1.5°C and +3 °C), ecosystem zone (pioneer zone, low marsh, and high marsh) , and soil depth (within subject / repeated measure) on TBI parameters for each year separately. Because we were not primarily interested in year-to-year differences, ANOVAs were used separately for each year to better understand the interactions between warming, zone, and soil depth."*

Furthermore, there is no mention of analysis done to produce figure 5 and it is hardly discussed in the results. Lacking this information it is hard to properly assess the results.

The temperature-induced change in k and S (deltak)was calculated relative to the ambient controls (k0) for each marsh zone separately: $\Delta k$ (%) = $(k_t/k_0 - 1)$ x 100 and $\Delta S$ (%) = $(S_t/S_0 - 1)$ x 100, t represents the value under different warming treatments (i.e. +1.5 and +3.0 ° C). Linear regression was used to analyze the relationships of $\Delta k$ and $\Delta S$ with soil depth. This information have been added to the Methods (line 166-172) as well as to the caption of figure 5 (line 205-209) in the revised ms.

Personally, I think Figure 5 gives a much better look into how the different factors influence decomposition rate k and S. I think expanding on this analysis would improve the manuscript as it sheds more light what factors/conditions affect decomposition in salt marshes. This could also be a way Figure 3 and 4 could then potentially be moved to supplementary material.

We agree with the reviewer that there is some redundancy between the figures and that Figure 5 yields more information than Figures 3-4. We are okay with the idea of moving Figures 3-4 to the

Supplement, in case the editor support on this. For now, we kept the figures in the main part. Figures have been improved in accordance with other comments provided below.

I also wondered why the authors did not use the measured soil reduction index in their analysis, as they as they discuss the influence of redox a lot in the discussion, but have not directly tested these links in their analysis. Why not use reduction index as predictor of k and S?

We did not quantify soil reduction inside the experimental plots but along a transect directly adjacent to the plots. Thus, there are no data on warming effects on soil reduction. Soil reduction was assessed to characterize the three marsh zones across which the experimental plots are distributed, i.e., pioneer zone, low marsh, high marsh. We originally intended to present this information in the Methods section just like we provided other aspects of the site and zone characterization here (e.g. plant-community composition, surface elevation) but were asked by the editor to move it to the results section prior to peer review. In any case, we believe these are important data to show that soil redox conditions actually differ between marsh zones and with respect to soil depth. However, we cannot use these data for e.g. correlations with S or k.

Specific comments

line 68. What do the authors mean with short- and mid-term warming?

Short- and mid-term was intended to refer to year 1 and 2 of the experiment, respectively. However we are not using this concept in the later parts of the ms and have removed this and related statements (line 70-71).

Line 96 Can the authors also indicate the location of the different warming treatments. And were these treatments randomly assigned?

Treatments were randomly assigned. Locations have been added to the figure 1.

[Figure]

Line 132. Not clear if it is 1 PVC stick per zone or whether there is replication? The transect in the next line adds to my confusion. Figure 6 legend states "n = 6 observations per zone, deployed over four consecutive deployment campaigns (July-October)". Clarify in methods.

We improved the description for the revised version (line 145-160).

*There were 6 sticks per zone (n = 6), and 18 (N = 6 x 3) sticks per campaign, in total N = 72 (n = 4 campaigns x 18) sticks were analyzed. These measurements were conducted along a transect directly adjacent to the experimental plots. The Reduction Index describes the fraction of $FeCl_3$ paint that is removed from the PVC stick after four weeks of deployment in the field. The IRIS method utilizes the property of the ferrihydrite paint to be reduced from solid-phase Fe(III) to soluble Fe(II) under anoxic soil conditions and in presence of microbial Fe-reducers. The area of removed paint from PVC sticks is used as a proxy for soil reduction (Reduction Index). Upon the 4-wk deployment phase, sticks were removed from the soil and cleaned carefully with tap water to remove soil particles. Each stick was scanned to create digital images for further processing. Image analysis was conducted applying a supervised classification on randomly chosen sticks from different field campaigns. Classification was done using the software ArcGIS Pro. In total, 4300 points were classified as either reduced, not reduced or errors (background, scanning effects). RGB color values (0-255) of classified points were retrieved using the Extract Multi values to Points function. The classification was included in a Random Forest model (confusion matrix 1.5%) using the software R. This model allowed for a pixel-wise classification of the scanned IRIS sticks. Sticks were analyzed in increments of 5 cm, covering a depth gradient from 0 to 30 cm. Reduction Index was calculated as an unitless value ranging from 0 to 1 based on the share of reduced pixels from the total pixels.*

Figure 3 Hard to read this figure as the error bars of the different treatments are overlapping. Please adjust figure so it is possible to discern the different treatments per soil depth.

Figure 3 and Figure 4 have been improved and used the three different error styles (i.e wide, medium, and small bars) to distinguished treatments.

Line 114 Incubation period different in 2018 and 2019, June-Sept vs May-July. Why? This does explain why temperatures were higher in 2018 as it is later in the season.

The two incubation experiments were started after the warming treatment was switched on in each year: In year 1, warming was switched on in May, in year 2, warming started in April. We were not primarily interested in year-to-year differences, but repeated the incubation in order to test if results from year 1 are replicable We agree with the reviewer and have added this aspect to the Discussion section.

Line 150-153 It is not clear to me what the authors are trying to say. Clarify

It has been changed accordingly.

Figure 3c Why high marsh higher k in +1.5 vs +3.0 treatment?

This is indeed an interesting pattern. If higher k at 1.5 vs 3.0 was only observed at the soil surface, drought effects could have played a role in the less-frequently flooded high marsh. However, the pattern is relatively consistent throughout the soil profile. We elaborated on this point in the revised Discussion section (line 273-278), focusing on plant-microbe interactions and nutrient constraints as potential drivers of the effect (compare Noyce et al. 2019). It is also important to stress that the pattern disappears in year 2.

*Our data suggest that warming effects on litter breakdown are not necessarily linear. In year 1 of the experiment, 1.5°C warming had a stronger positive effect on the initial decomposition rate than 3.0°C warming in the high marsh. We doubt that potential warming-induced drought effects in the soil surface were the responsible driver of this pattern, because higher k under +1.5°C vs. +3.0°C warming was relatively consistent throughout the 60-cm soil profile. Instead, it is possible that plant-microbe interactions caused the observed pattern. For instance, warming-stimulated nutrient demands and belowground trait responses in plants could have negatively affected microbial communities via competition for nutrients (Noyce et al., 2019).*

Why k higher in warming treatments in 2019 vs 2018? Bigger difference in temperature?

Temperature differences were consistent between years: The mean delta temperature across all marsh zones and plots ranged from 0.99° to 1.66° C for the +1.5°C treatment, and 1.61° to 2.81° C for the +3.0°C treatment in 2018, and the mean delta temperature across all marsh zones and plots ranged in 2019 from 1.26° to 1.82° C for the +1.5°C treatment, and 2.06° to 2.81° C for the +3.0°C treatment (Figure S1). One potential reason is greater microbial biomass in year 2 vs. 1 of the experiment as warming stimulated plant growth and substrate input to the soil system. However, also differences in the actual temperature (not delta T) and the slight seasonal shift between the two incubation periods could have affected the magnitude of warming effects. We have discussed this point in the revised Discussion (line 280-287).

[Figure]

Figure S1 Absolute soil temperature across marsh zones and warming treatments during the incubation phases of year 1 and year 2.

Technical corrections:

Line 49 "in" question

It has been changed accordingly.

Line 50. Do the authors mean labor intensive instead of efficient? Constructing a lot of litter bags is labor intensive in my opinion.

It has been changed accordingly.

Line 54 rephrase: represents a widely used standardized litter bag approach

It has been changed accordingly.

Line 68 replace "over" with "for". Over implies the incubation time was either 1 or 2 years, but there were separate 3-month incubations in each year.

It has been changed accordingly.

Line 81 Can the authors use a more widely known standard like meter above sea level for to indicate elevation instead of NHN?

Meter above sea level in Germany is referenced against NHN, i.e. 1 m NHN = 1 m above sea level. Made clear now in ms.

Line 132 "." After citation

It has been changed accordingly.

Line 132 remove "from pioneer…. High marsh."

It has been changed accordingly.

Line 248 known

It has been changed accordingly.

**Referee #3**

This paper examines patterns of initial decomposition rate and stabilization factor over a flooding gradient and depth profile in a tidal marsh. The focus on decomposition over an elevation gradient and belowground, where the greatest contribution by plants to blue C accumulation occurs, represents an important contribution to the literature. Further, the use of standard substrate to control for litter quality allows for a focus on abiotic drivers of decay. Overall, this is a good paper that, with some clarifications to the methods and stats and additional interpretation of results, will add valuable insights to decomposition processes in tidal marshes that are especially vulnerable to climate change.

Specific comments and questions:

L60-61: clarify by explaining what their proposed mechanism is for how the lack of oxygen inhibits warming effects.

We provided more background here, focusing on the enzymic latch hypothesis and the recalcitrance of chemically stable tissues, such as phenolic compounds, in the absence of oxygen (line 61-63).

*"The authors suggest that the absence of oxygen can inhibit warming effects on soil microbial activity because phenolic compounds accumulate under anoxic conditions and inhibit microbial hydrolytic enzyme activity via the "enzymic latch" mechanism."*

L63: reiterate that the use of a standard substrate is needed to understand warming effects.

*It has been changed accordingly.*

L70: is the first hypothesis expected regardless of soil depth?

*Yes, the first hypothesis is meant to be general, and the second hypothesis is meant to further specify and explore the interaction effects with marsh zone and soil depth.*

Site description – Figure 1 is difficult to see, and the zones are not clearly defined in the text. It would be helpful to explain how the zones are oriented relative to the open water, with the pioneer zone along the shoreline and the high marsh farthest inland. Also, how is "pioneer zone" defined?

*The figure 1 have been improved accordingly, and more detail have been added to the site description.*

[Figure]

*"The pioneer zone is a typical feature of NW European salt marshes and is defined as the area where pioneer vegetation covers ⩾ 5 % (Peterson et al., 2014). In the Wadden Sea region its average surface elevation is below mean high tide. Thus, the pioneer zone is typically flooded twice daily (Esselink, 2017).".*

Experimental design – How was the soil warming established and verified along the soil depth gradient? Was there uniform warming of the soil column or did it vary with depth? It would be nice to see a

graph of these data. Did you confirm treatment conditions of +1.5 and +3 degrees warming? Why was the average soil temperature from -10 and -60 cm used as opposed to looking at temperature along the soil depth gradient at the same intervals at which decomposition and soil reduction were measured?

More detail is provided to a similar comment by Reviewer #1 and this information has been added in the Method part (line 106-111).

*"Belowground temperature was monitored continuously and logged at 5-min intervals using custom made thermistors and dataloggers. To control the heating rate evenly throughout the soil profile, sensors were placed at -5, -25, and -75 cm depth below the soil surface. At -5 cm, the highest variation in mean delta temperature across all marsh zones and plots ranged from 1.43° to 1.67° C for the +1.5° treatment, and 2.54° to 2.99° C for the +3.0° treatment. At -25 cm depth, the mean delta values ranged from 1.51° to 1.55° C for the +1.5° treatment and 2.87° to 3.02° for the +3.0° treatment. At -75 cm, delta values ranged from 1.14° to 1.43° C for the +1.5° treatment and 1.92° to 2.36° for the +3.0° treatment (Rich et al. under review)."*

Decomposition – why was this examined across two different periods (June-Sept vs. May-July) in the two years? It is not surprising that ambient temperatures were cooler in year 2 (late spring/early summer) than in year 1 (late summer), which may have contributed to the larger effect sizes of warming in year 2 compared to year 1. Address why these time periods were selected, and later discuss how this could have affected results.

The two incubation experiments were started after the warming treatment was switched on in each year: In year 1, warming was switched on in May, in year 2, warming started in April. We were not primarily interested in year-to-year differences, but repeated the incubation in order to test if results from year 1 are replicable. We have improved this clarity in the discussion part (line 280-287).

*"While qualitatively, warming effects on k and S were often similar between year 1 and 2, quantitatively, effects were generally much more pronounced in year 2 than 1 (Figures 3-4). The experimentally achieved delta temperature values were consistent between years (Table S1); however, differences in the actual temperature (not delta T) and the seasonal shift of 13 weeks between the two incubation periods of year 1 and 2 deployment phases could have affected the magnitude of warming effects. Absolute soil temperatures were lower in year 2 than 1 (Figure S1), which could have resulted in the amplified warming effect. It is also possible that changes in the microbial community with increased treatment duration and/or greater microbial biomass as warming stimulated plant growth and substrate input to the soil system contributed to the observed effect amplification over time."*

Statistics – This section is lacking details and does not fully track with the results presented. Were years compared statistically or tested separately? Why or why not? How were the effect sizes determined and analyzed, and why was this only examined in year 2? How was soil reduction analyzed?

We used the two-way repeated-measures ANOVA to test the effects of warming, zone, and depth (depth =within subject / repeated measure) for each year separately.

We were not primarily interested in year-to-year differences and therefore used ANOVAs separately for each year to better understand the interactions between warming, zone, and soil depth.

Based on the results of two-way repeated-measure ANOVA, the significant warming x depth x zone interaction was detected in year 2 only. A more detailed presentation of effect sizes was therefore restricted to year 2.

We improved clarity of the statistical methods section in the revised ms (line 163-166).

Soil reduction: this parameter was assessed using the Indicator of Reduction in Soils technique. We agree that the information given on the methods is insufficient. We provided a thorough description in the revised Methods section (line 147-160).

*"There were 6 sticks per zone (n = 6), and 18 (N = 6 x 3) sticks per campaign, in total N = 72 (n = 4 campaigns x 18) sticks were analyzed. These measurements were conducted along a transect directly adjacent to the experimental plots. The Reduction Index describes the fraction of $FeCl_3$ paint that is removed from the PVC stick after four weeks of deployment in the field. The IRIS method utilizes the property of the ferrihydrite paint to be reduced from solid-phase Fe(III) to soluble Fe(II) under anoxic soil conditions and in presence of microbial Fe-reducers. The area of removed paint from PVC sticks is used as a proxy for soil reduction (Reduction Index). Upon the 4-wk deployment phase, sticks were removed from the soil and cleaned carefully with tap water to remove soil particles. Each stick was scanned to create digital images for further processing. Image analysis was conducted applying a supervised classification on randomly chosen sticks from different field campaigns. Classification was done using the software ArcGIS Pro. In total, 4300 points were classified as either reduced, not reduced or errors (background, scanning effects). RGB color values (0-255) of classified points were retrieved using the Extract Multi values to Points function. The classification was included in a Random Forest model (confusion matrix 1.5%) using the software R. This model allowed for a pixel-wise classification of the scanned IRIS sticks. Sticks were analyzed in increments of 5 cm, covering a depth gradient from 0 to 30 cm. Reduction Index was calculated as an unitless value ranging from 0 to 1 based on the share of reduced pixels from the total pixels."*

For the discussion and methodological considerations, how much could leaching be contributing to the results and different findings for k and S along the flooding gradient? How did the PVC influence the hydrology or connectivity of the tea bags with their surroundings? Was the temperature gradient verified within those PVC pipes? It would also be useful to revisit the importance of litter quality, as well as species-specific differences in decay with species turnover along the elevation gradient. While this study was designed to avoid plant influences, brief discussion of how it could affect these patterns, and how shifts in community composition with sea-level rise is another climate change driver to be considered that, if species differ in their contributions to blue C, could have implications for marsh resilience.

Similar comments with respect to leaching, PVC-post effects on the abiotic environment, and litter quality were made by R1 and R2 (compare above). The section on Methodological considerations have been improved in accordance.

For leaching: *"A number of recent studies have highlighted the importance of leaching in the context of the TBI (Gessner et al., 2010; Lind et al., 2022; Marley et al., 2019). Because leaching is a rapid process, particularly in wetlands, we assume that leaching was complete throughout all vegetation*

*zones and thus, did not contribute to the observed variability in k and S. Indeed, k did not increase, and S did not decrease with flooding (elevation gradient) or soil moisture (depth gradient) suggesting that leaching did not (overly) control our results." (line 314-318)*

For PVC: *"The use of PVC posts may have affected drainage and thus redox conditions of the deployed litter materials after flooding events. This could have amplified the redox differences between frequently and rarely flooded vegetation zones we observed. We argue that this potential effect on drainage is unimportant for the interpretation of our results, because our study was primarily designed to gain mechanistic insight, not to capture actual rates of litter breakdown." (line 290-293)*

For litter quality: *"Warming and other climate change drivers are expected to induce changes in the quality of plant litter and other organic matter inputs accumulating in salt-marsh soils, for instance through shifts in the plant community composition that can potentially counterbalance or amplify the effects on decomposition processes described here (Mueller et al. 2018; 2020). Future research within the MERIT project will therefore address litter quality-feedback effects on decomposition processes in order to gain a more complete understanding of warming effects on salt-marsh soil carbon cycling." (line 300-304)*

Technical comments:

L13: clarify "plant production"

It has been changed accordingly.

L15: suggest "entire intertidal flooding gradient"

It has been changed accordingly.

L17: delete "of" before "(k)"

It has been changed accordingly.

L54: delete "probably"

It has been changed accordingly.

L59: offset "and thus strongly reducing" with commas

It has been changed accordingly.

L69: what is short- and mid-term warming effects? Is this in reference to projected warming of +1.5 vs. 3 degrees?

It has been changed accordingly.

L71: combine sentences so that it reads "…soil, and (2) that warming…"

It has been changed accordingly.

L77: "has operated" instead of "operates"

It has been changed accordingly.

L79: change comma after climate to semicolon

It has been changed accordingly.

L148: should this be "appear to be consistent"?

It has been changed accordingly.

L150: this is unclear. What do you mean by "refer the significant interaction"?

It has been changed accordingly.

L152: clarify that the relationship is "with increasing soil depth"

It has been changed accordingly.

L192: change to "a large" instead of "an"

It has been changed accordingly.

L209: add a comma after the citation

It has been changed accordingly.

L248: "known"

It has been changed accordingly.

---

## Author Response (AR3)

We would like to thank the reviewer for his/her time and constructive comments. Below we respond to each comment separately (in blue font) referring to the line numbers of the original submission.

General comments

The authors have addressed the comments of the reviewers and provided the needed additional information regarding their experimental design, methods used and methodological considerations. As a result, the manuscript has much improved. However, I still have some minor comments/edits.

Specific comments

I suggest making two paragraphs in the statistical analysis section to separate the ANOVA analysis and linear regression analysis. Move the text about pairwise comparison and testing ANOVA assumptions after explanation of ANOVA analysis.

We agree and have re-ordered the paragraph in statistical analyses section (line 165-171).

Figure 3 is still hard to read because of overlapping points and error bars (dodge the points so they do not overlap). Could the authors adjust the figure.

Figure 3 and Figure 4 have been improved.

[Figure]

The additional text for Figure 5 is redundant and can be removed as it is already described in the methods.

It has been changed accordingly.

Technical corrections:

Line 74 "to stabilize"

It has been changed accordingly.

Line 296 delete second quantitatively

It has been changed accordingly.

Line 308 delete not critical

It has been changed accordingly.

Line 334 "affect our results"

It has been changed accordingly.

Additional comments (line numbers relative to tack changes version (bg-2022-189-ATC3.pdf):

L96: replace 'Del'13C signature by the symbol of even better by 13C signature , with '13' superscript

It has been changed accordingly.

L117: avoid formulation "delta temperature" either define and use symbol or use "temperature difference"

It has been changed accordingly.

L297: avoid formulation "delta temperature values" either define and use symbol or use "temperature difference".

avoid formulation "year 2 than 1" use "in year 2 compared to year 1" - or just "in year 2" where appropriate.

It has been changed accordingly.

L308 rephrase "not critical unimportant"

It has been changed accordingly.

L309 replace "mechanistic" by "qualitative" (if the difference is that you are not interested in accurate numbers)

It has been changed accordingly.

L330-335: tell what is meant with leaching (DIC, DOC, POM, POC, other?) and why "vegetation zones ". Which variability in k and S do you refer to (their or your study)?

We improved this paragraph in the revised ms (line 310-315).

Conclusions: generally avoid symbols and abbreviations in a way that the paragraph can be understood without reading the main text.

It has been changed accordingly.

L339: spell out OM, k and S

It has been changed accordingly.